# Progesterone Can Directly Inhibit the Life Activities of *Toxoplasma gondii* In Vitro through the Progesterone Receptor Membrane Component (PGRMC)

**DOI:** 10.3390/ijms23073843

**Published:** 2022-03-31

**Authors:** Yihan Wu, Xiao Zhang, Yong Fu, Jing Liu, Yangfei Xue, Qun Liu

**Affiliations:** 1National Animal Protozoa Laboratory, College of Veterinary Medicine, China Agricultural University, Beijing 100193, China; yrxah1119@163.com (Y.W.); tzhangxiao@126.com (X.Z.); fuyongvet@163.com (Y.F.); liujingvet@cau.edu.cn (J.L.); 17835422205@163.com (Y.X.); 2Key Laboratory of Animal Epidemiology of the Ministry of Agriculture, College of Veterinary Medicine, China Agricultural University, Beijing 100193, China; 3Department of Preventive Veterinary Medicine, College of Veterinary Medicine, Shandong Agricultural University, Tai’an 271000, China

**Keywords:** *Toxoplasma gondii*, progesterone (P4), pregnancy, progesterone membrane receptor protein (PGRMC)

## Abstract

*Toxoplasma gondii (T. gondii)*, as an opportunistic pathogen, has special pathogenic effects on pregnant animals and humans. Progesterone (P4) is a critical hormone that supports pregnancy, and its levels fluctuate naturally during early pregnancy. However, little is known about the association of host P4 levels with the infectivity and pathogenicity of *T. gondii*. Our study showed that P4 significantly inhibited the invasion and proliferation of tachyzoites, resulting in abnormal cytoskeletal daughter budding and subsequent autophagy in vitro. To investigate the underlying mechanism, we identified a *Toxoplasma gondii* progesterone membrane receptor protein (TgPGRMC) that was localized to the mitochondrion and closely related to the effect of P4 on tachyzoites. The knockout of the *pgrmc* gene conferred resistance to P4 inhibitory effects. Our results prove the direct relationship between P4 single factors and *T. gondii* in vitro and demonstrate that TgPGRMC is an important link between *T. gondii* and P4, providing a new direction for research on *T. gondii* infection during pregnancy.

## 1. Introduction

*Toxoplasma gondii* is an obligate intracellular parasitic protozoan that can parasitize the nucleated cells of almost all warm-blooded animals and is a serious zoonotic pathogen. Animals and humans during pregnancy are more susceptible to *T. gondii* because it is an opportunistic parasite [1,2,3]. Significantly increased infection rates during host pregnancy were observed, and reproductive obstacles such as abortion, stillbirth, weak fetus, and the congenital infection of the fetus were caused by *T. gondii* [4,5]. Therefore, serious *T. gondii* infections during pregnancy have been a research hotspot over the years.

Progesterone is a fluctuated physiological factor during pregnancy in animals and humans, and the concentration of progesterone in the serum of pregnant women is over 400–600 times higher than that during the follicular phase [6,7,8]. We hypothesized that changes in progesterone concentration during pregnancy may be related to an increase in the *T. gondii* infection rate during pregnancy.

The relationship between progesterone and pathogens is complicated. It is currently known that progesterone can affect the infection of parasites by modulating innate and acquired immune responses [9,10,11]. In order to create a suitable environment for the growth and development of the embryo, sex hormones need to regulate the host’s immunity, reduce the immune rejection of the fetus, and also provide favorable conditions for the invasion of pathogens. During pregnancy, the organism’s bias toward an immune response is mediated by type II helper T cells (Th2) [12,13]. Progesterone can promote the expansion of Th2-type cells and induce the production of IL-4, and IL-4 can reduce the damage to maternal trophoblasts by the immune rejection reaction induced by the paternal antigens. At present, it is believed that the organism prevents the infection of *T. gondii*, which is mainly dependent on innate immunity and IFN-Y produced by acquired immunity. Progesterone can induce the production of some cell regulatory factors (such as IL-4) to inhibit the production of IFN-γ and the activation of the NF-kB pathway, which can reduce the tolerance of pregnant mice to *T. gondii* [14]. Thouvenin and colleagues found that pregnant IL4−/− mice showed lower susceptibility to *T. gondii* infection and a lower materno-fetal transmission rate [15]. Luft and Remington found that pregnant mice had significantly lower NK cell-killing activity than virgin mice when infected with *T**. gondii* [16]. These research results indirectly suggest that sex hormones during pregnancy can regulate some immune cells and factors, affect host immunity, and ultimately change the tolerance to *T. gondii.* In addition, progesterone can induce an increased production of asymmetric antibodies and inhibit the cytotoxicity of natural killer (NK) cells, eventually leading to an increase in the pathogen invasion rate [14,17].

Sex hormones during pregnancy can modulate the host’s immunity to change the pathogenicity of the pathogen; additionally, they can also directly act on pathogens and induce pathogenic changes. The direct effects of progesterone on the growth and metabolism of parasites have been reported. Lingnau and colleagues found that progesterone promoted the proliferation of *Plasmodium falciparum* gametocytes in vitro [18].

Whether P4 can directly act on *T. gondii* to regulate its growth and reproduction has not been reported. Our study focused on the role of progesterone on *T. gondii* survival and pathogenicity, and we tried to explore the possibility of the existence of *T. gondii* progesterone receptors to further study the relationship between *T. gondii* and progesterone.

## 2. Results

### 2.1. Progesterone Inhibits the Invasion of T. gondii

To verify the influence of P4 on *T. gondii* survival in vitro, we performed invasion assays using serum-free medium with various concentrations of progesterone, and the percentages of parasite-infected cells were monitored. Compared with the control group, a general invasion inhibition was observed in progesterone-treated groups (Figure 1a). To verify the comprehensive influence of P4 on *T. gondii* proliferation, we performed invasion assays using a serum-free medium with various concentrations of progesterone. Plaque assays showed a significant reduction in plaque formation in progesterone treatment groups, especially at 10^−5^ M progesterone; parasites hardly formed plaques visible to the naked eye within 7 days, which indicates that progesterone inhibits the survival of *T. gondii* in vitro (Figure 1b,c).

### 2.2. Progesterone Affects the Morphology and Division of T. gondii and Induces Autophagy

To study the effect of P4 on the development and morphology of tachyzoites, we continued to cultivate tachyzoites in the progesterone environment and then measured the morphological changes through IFA. SAG1 was used to mark the morphology of *T. gondii*. After a 48 h culture in serum-free medium containing progesterone, the morphology of tachyzoites was abnormal; a large vesicle-like structure was formed in the vacuole, and the boundary of the parasitophorous vacuole became blurred (Figure 2a). At the same time, we found that progesterone can induce abnormal cytoskeletal daughter budding. The inner membrane complex protein 1 (IMC1) was used to mark the cytoskeletal daughter budding [19]. We used IMC1 as the marker to determine the affection of progesterone on tachyzoite division. The results showed that cell division in the same parasitophorous vacuole was asynchronous in the presence of progesterone (Figure 2c). In addition, the apicoplast (ACP), whose division occurs concurrently with nuclear division and daughter cell budding, showed abnormal division after a 48 h culture in serum-free medium containing progesterone [20] (Figure 2e). We performed a statistical analysis on the proportion of *T. gondii* with abnormal morphologies. One hundred randomly selected vacuoles from two independent experiments were quantified. As the concentration of progesterone increased, the proportion of abnormalities continued to increase (Figure 2b,d,f). 

After this, a flow cytometry analysis of *T. gondii* tachyzoites stained for fragmented DNA using a TUNEL assay was performed. Parasites showed 32.4%, 51.3%, and 94.3% TUNEL positive rates when treated with 10^−7^ M, 10^−6^ M, and 10^−5^ M progesterone for 3 h, respectively, which are significantly different to the control ~27.5% TUNEL positive rates (Figure 3a). We also detected the expression of ATG8 under the action of progesterone. ATG8, an autophagy-related protein, was the marker of *T. gondii* autophagy [21]. A dot accumulation of ATG8 in the cytoplasm was observed under 10^−5^ M progesterone (Figure 3b). We observed the effect of progesterone on the ultrastructure of *T. gondii* by using transmission electron microscopy. The results showed that, under normal circumstances, RHΔ*ku80* tachyzoites contained well-organized organelles. By contrast, RHΔ*ku80* tachyzoites were often deformed, vacuolized, and showed many autophagic vacuoles emerging in the cytoplasm after 3 hours of 10^−5^ M progesterone treatment [22] (Figure 3c).

### 2.3. Progesterone Membrane Receptor in T. gondii

Progesterone can directly act on tachyzoites and affect their life activities, suggesting that *T. gondii* may have effector molecules that interact with progesterone. The hormone receptors of many organisms have relatively conservative structures. We conducted a bioinformatics analysis of the whole genome of *T. gondii* and identified that there is a gene similar to the mammalian progesterone membrane receptor PGRMC, which we called TgPGRMC (TGGT1_276990). The motif of this gene was analyzed using Motif Scan (Motif Scan (sib.swiss), accessed on 9 January 2019), and it was found to have a conserved hormone-binding site, six CK2 phosphorylation sites, three PKC phosphorylation sites, and four myristyl sites (Figure 4a). A phylogenetic analysis of the identified proteins showed that the gene is relatively conserved in the parasitic protozoa and has high homology with the hormone-binding proteins of some sporidiosis pathogens (Figure 4b).

After this, we used the method of molecular modeling to study the binding mode of progesterone and the possible three-dimensional structure of TgPGRMC and used the docking analysis to determine whether this protein is the progesterone membrane receptor of *T. gondii*. The docking results showed that TgPGRMC has a progesterone-binding pocket, and 151A is likely to be the key residue for progesterone to bind to the target protein (Figure 4c,d).

### 2.4. TgPGRMC Localizes in the Mitochondria and Is Not Necessary for the Parasite Lytic Cycle

To determine the localization of TgPGRMC, we first attempted to tag the *pgrmc* gene with three epitope HA tags at the C-terminal of the genomic locus (Figure 5a), which were verified by Western blot (Figure 5b). IFA assays show that TgPGRMC is located in the mitochondria (Figure 5c).

To further investigate the biological role of TgPGRMC, we used CRISPR-Cas9 technology to successfully construct *pgrmc* gene knockout strain (Figure 5d). The complete *pgrmc* gene deficiency parasites were validated by PCR (Figure 5e). We monitored the formation of plaques by continuous seven-day culture, and the plaque assay showed there was no obvious difference between RHΔ*ku80* strain and *pgrmc* gene knockout strain (Figure 5f,g).

### 2.5. Knockout pgrmc Gene Alleviates the Inhibitory Effect of Progesterone on T. gondii

To study the relationship between TgPGRMC and progesterone, we used the gene knockout strain and the wild-type strain to conduct a plaque assay to detect the changes in their growth and proliferation ability under progesterone stimulation. The results showed that the knockout of the *pgrmc* gene conferred resistance to 10^−6^ M progesterone inhibitory effects. However, under 10^−5^ M progesterone, *pgrmc* gene knockout strain, such as wild-type strain, cannot form visible plaques (Figure 6a,b).

For further evaluation of the effects of the *pgrmc* gene on *T. gondii* pathogenicity, we intraperitoneally inoculated 100 RHΔ*ku80* and Δ*pgrmc* tachyzoites to evaluate the virulence in BALB/c mice (five mice per strain). The survival curve showed the pathogenicity of the Δ*pgrmc* strain to mice was not much different from that of the RHΔ*ku80* strain, and all mice died within 7 days (Figure 6c).

## 3. Discussion

Pregnant women and pregnant animals are more likely to be infected with *T. gondii* because it is an opportunistic parasite. As one of the physiological factors with the greatest variation during pregnancy, progesterone levels increase significantly during pregnancy, and we speculate that progesterone may directly regulate the growth and metabolism of *T. gondii*, making it easier for the parasite to invade the host and multiply rapidly, forming a highly pathogenic situation. Interestingly, the results show that the direct progesterone treatment of *T. gondii* inhibits its invasion and induces autophagy. However, the concentrations of several hormones, including estradiol, progesterone, testosterone, and cortisol, increase substantially during pregnancy [23,24]. There may be other factors that act synergistically with progesterone, and progesterone may not be the leading factor affecting the high morbidity of *T. gondii* during pregnancy. Our previous research found that estradiol promoted Pru (Type II) and VEG (Type III) infection and thus significantly contributed to the pathogenicity of *T. gondii* in mice [4]. In this study, we prove that progesterone can inhibit the life activity of *T. gondii* under a single action condition in vitro, though these findings cannot represent the result in vivo. The relationship between P4 and *T. gondii* in vivo remains to be studied.

At present, it is widely believed that progesterone can directly regulate and promote the growth, proliferation, and differentiation of organisms through binding with progesterone receptors. Since the discovery of the first progesterone-related receptor in 1980, the process by which progesterone regulates various life activities has been studied in depth. The receptor, later known as the progesterone receptor (PGR), is located in the nucleus and regulates gene transcription [25]. Since then, two other putative progesterone receptor families have been identified; they are the AdipoQ receptor family (PAQR) and the progesterone membrane receptor family (PGRMC) [26,27]. The main functions of these two receptor families are focused on initiating “nongenomic” effects [28]. In most organisms, progesterone receptors are highly conserved. However, we have conducted much work and have not found that there is a nuclear receptor similar to PGR in *T. gondii*. Therefore, we focused on the “nongenomic effects” mediated by progesterone and found the TgPGRMC studied in this article. The function of the gene was explained. The knockout of the *pgrmc* gene does not affect the survival of *T. gondii* in vitro but makes it more viable in a low-concentration progesterone environment (10^−6^ M). However, Δ*pgrmc* strain has no resistance to the stimulation of 10^−5^ M progesterone. We hypothesized that although TgPGRMC is related to progesterone, there should be other proteins in *T. gondii* involved in the regulation of this process.

No significant difference was observed in the pathogenicity to mice between the RHΔ*ku80* and Δ*pgrmc* strains. This may be due to the fact that the type I RH strain is too virulent for mice infection studies. Moreover, the content of progesterone in nonpregnant mice may not meet our experimental requirements.

Intriguingly, TgPGRMC is located in mitochondria. Progesterone is a steroid hormone, and cholesterol import into mitochondria is essential for steroidogenesis [29]. Steroid hormones will, in turn, regulate the number of mitochondria in the cell [30]. We suspect that the mitochondrial location of TgPGRMC indicates that progesterone can affect mitochondrial functions and thus affect the life activities of *T. gondii*. However, our previous research showed that progesterone has no effect on reactive oxygen species (ROS) production in *T. gondii*, which is closely related to mitochondria [31].

How P4 activates TgPGRMC to affect the transcriptional expression of some genes is not yet fully understood. It is generally believed that progesterone first inhibits the activity of Tcf/Lef through TgPGRMC, and the change in Tcf/Lef activity will regulate the expression of many target genes, affecting cell survival and cell proliferation [32]. However, whether there is a Tcf/Lef pathway in *T. gondii* and in the relationship between TgPGRMC and Tcf/Lef is still unknown. More in-depth work is needed to verify and discuss these remaining questions.

## 4. Materials and Methods

### 4.1. Ethics Approval and Consent to Participate

All experiments with mice were performed in strict accordance with the recommendations of the Guide for the Care and Use of Laboratory Animals of the Ministry of Science and Technology of China. All procedures were approved by the Institutional Animal Care and Use Committee of China Agricultural University (under the certificate of Beijing Laboratory Animal employee ID CAU20161210-2).

### 4.2. Progesterone Treatment

Progesterone (Sigma, P0130, St. Louis, MO, USA) was first dissolved in dimethyl sulfoxide (DMSO, Invitrogen, D12345, USA) (10^−5^ M) and stored at −20 °C until use. Progesterone was tested at a final concentration range of 10^−7^ M to 10^−5^ M, with equivalent concentrations of DMSO or serum-free medium used as controls in each experiment. For P4 treatment, cells were first seeded in a 12-well plate in DMEM with 2% FBS but without phenol red. Cells were left to adhere for 24 h, after which the medium was replaced by serum-free medium or progesterone-containing medium.

### 4.3. Parasite and Culture

RHΔ*ku80* strain was used as the parental parasites to construct TgPGRMC knockout strain. *T. gondii* tachyzoites were cultured in HFFs (human foreskin fibroblasts, Cell Bank of the Chinese Academy of Sciences, Shanghai, China) and continuously passaged in DMEM (Macgene, CM17206, Beijing, China) containing 2% fetal bovine serum (FBS) at 37 °C and 5% CO_2_ in a humidified incubator. In addition, the medium was replaced 24 h after inoculation. Transfections were carried out by electroporation using 10^7^ freshly egressed or mechanically released parasites as previously described [33].

### 4.4. Plaque Assays

The plaque experiment was performed using HFF cells cultured in 12-well cell culture plates. When HFFs grew to 80–90%, the medium was replaced with fresh medium containing 2% FBS. Freshly released tachyzoites were collected from the cells for purification and counting, with approximately 150 tachyzoites inoculated per well and at least two replicates in each group, and were incubated in a constant temperature incubator at 37 °C and 5% CO_2_ for 6–7 days. Then, the medium was removed, and the cells were washed with PBS twice. The cells were then fixed with 4% paraformaldehyde for 30 min and stained in crystal violet solution for 1 h. The results were visualized by microscopy using image acquisition and plaque area measurement as described by others [34].

### 4.5. Invasion Assay

For the invasion assay, HFFs growing in 12-well plates seeded on coverslips were inoculated with 1 × 10^5^ tachyzoites and incubated for 30 min. After the cells were washed with PBS three to five times, fresh medium was added and incubated for another 20 h. For IFA experiments, rabbit TgGAP45 (1:200) was used to stain the parasite membrane, DAPI (1:100) was used to stain the nucleus, and the fluorescence microscope was used for observation. The percent of invasion was represented as the number of vacuoles per host cell. Three independent experiments were performed.

### 4.6. Mice and Experimental Infections

BALB/c mice aged 6 weeks were purchased from Charles River (Beijing). Adequate drinking and eating were ensured, and the animals were kept under 14 h of light every day. All mouse infection experiments described in this paper used freshly released tachyzoites resuspended in PBS, and the mice were infected by intraperitoneal injection.

### 4.7. Molecular Docking Test

The online website for SWISS-MODEL (expasy.org, accessed on 1 March 2020) was used to predict the three-dimensional structure of TgPGRMC; we used it to analyze the function and key active site residues and used the 3D molecular modeling software PyMOL to display the structure. The online website for LSCF Bioinformatics—Protein Structure—Binding Site (weizmann.ac.il, accessed on 26 March 2020) was used to identify the small-molecule pocket positions. The three-dimensional structure of progesterone was downloaded from PubChem. The molecular docking software AutoDock Vina was used to dock the three-dimensional structure of TgPGRMC with progesterone. The ligand conformation, orientation, position, and energy were scored, and the results were sorted. The lower the free energy is, the higher the probability of conformation, and the ligand–receptor binding conformation must be a low-energy conformation.

### 4.8. TUNEL Assay

TUNEL assays were performed using an apoptosis detection kit (Vazyme Biotech) as previously reported [35]. Freshly egressed tachyzoites were treated with P4 for 3 h, and medium with the same final concentration of DMSO was used as control. After fixation and permeabilization, parasites were incubated in TUNEL reaction mix with TdT enzyme, and nuclei were stained with DAPI. We identified and quantified positive cells by flow cytometry.

### 4.9. Construction of Transgenic Parasite Lines

To obtain the TgPGRMC-HA parasites, we constructed pLIC-HA-DHFR-TgPGRMC plasmid to tag the *pgrmc* gene with three epitope HA tags at the C-terminal of the genomic locus. We regarded 42 bp fragments upstream of the TgPGRMC translation stop codon as the 5′ flank homologous arm and 42 bp fragments downstream of the gRNA as the 3′ flank. The pLIC-HA-DHFR-TgPGRMC plasmid and the corresponding CRISPR-Cas9 plasmid were co-transfected into RHΔ*ku80* parasites and screened by pyrimethamine.

To construct Δ*pgrmc* strain, the E-CRISP sgRNA design website (http://www.e-crisp.org/E-CRISP/, accessed on 10 February 2019) was used, and CRISPR/CAS9 plasmids used for disrupting the *pgrmc* gene were constructed first. The pTCR-*pgrmc*-CD plasmids were constructed for homologous recombination. The 3′ and 5′ flanks of the *pgrmc* gene were cloned from the genomic DNA of RHΔ*ku80*. The chloramphenicol selection markers fused with the red fluorescent protein (CAT-RFP) cassette were cloned from the pTCR-CD plasmid. Then, the plasmids pTCR-*pgrmc*-CD and CRISPR/CAS9 were constructed by seamless cloning. The details of all the primers used in this paper are presented in Appendix A.

### 4.10. Transmission Electron Microscope

Samples were processed as described previously [36]. Briefly, HFF cells were infected with *T. gondii* for 12 h, cultured in serum-free medium containing progesterone for 3 h, and then were fixed in 1% glutaraldehyde (Polysciences Inc., Warrington, PA, USA) and 1% osmium tetroxide (Polysciences Inc., Warrington, PA, USA) in 50 mM phosphate buffer at 4 °C for 30 min. The ultrathin sections (70 nm thick) were sectioned with microtome (Leica M80), double-stained with uranyl acetate and lead citrate, and examined by a transmission electron microscope (Hitachi H-7500).

### 4.11. Immunofluorescence Assays and Western Blot

For immunofluorescence assays, HFFs were infected with parasites and incubated for 24 h; they were subsequently fixed with 4% paraformaldehyde (PFA) followed by treatment in 0.25% Triton X-100 and blocked with 3% of BSA. The cells were incubated with primary antibodies for 1 h, washed by PBS, and then incubated with secondary antibody for 1 h. Mouse anti-HA antibody was purchased from Sigma; mouse anti-SAG1, mouse anti-IMC1, mouse anti-ACP, and rabbit anti-GAP45 were maintained in our laboratory. Images were viewed using the Olympus IX70 Inverted Microscope and captured by Olympus DP Controller software.

For Western blotting, parasites were lysed in RIPA buffer followed by SDS-PAGE. The primary antibodies used were mouse anti-HA (1:5000, Sigma, St. Louis, MO, USA), mouse anti-Actin (1:5000), and mouse anti-ATG8 (1:100). Secondary antibodies used were goat anti-mouse or rabbit (1:5000, Invitrogen, Carlsbad, CA, USA).

### 4.12. Statistical Analysis

GraphPad Prism 6.0 was used to generate graphs and perform statistical analyses. All data were analyzed with the two-tailed Student’s *t*-test. The statistical analysis of survival curves was analyzed by “curve comparison”.

## Figures and Tables

**Figure 1 ijms-23-03843-f001:**
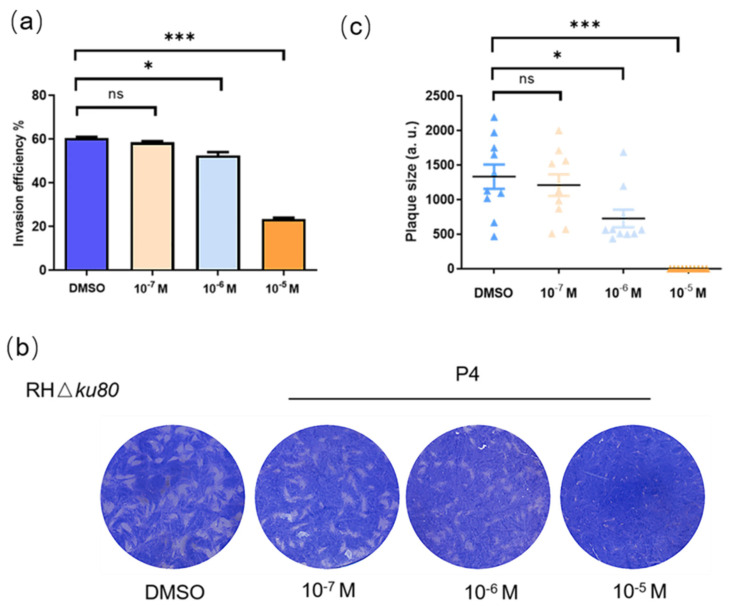
Percentage of infected cells after infection with tachyzoites: (**a**) Percentages of intracellular RHΔ*ku80* tachyzoites were calculated to determine the invasion efficiency in serum-free medium treated with progesterone at a concentration of 10^−7^ M, 10^−6^ M and 10^−5^ M. Control: medium with the same final concentration of DMSO (ns, not significantly, * *p* < 0.05, *** *p* < 0.001); (**b**) HFFs were infected with 150 tachyzoites, cultured with different concentrations of progesterone, and incubated for 7 days without any perturbation to allow plaque formation. The monolayer was stained with crystal violet. Control: medium with the same final concentration of DMSO; (**c**) Quantitative analysis of plaque area and number in the RHΔ*ku80* strain treated with 10^−7^ M, 10^−6^ M, and 10^−5^ M progesterone. The plaque formation in the progesterone treatment group was significantly inhibited with the increase in progesterone concentration. Plaque assays were carried out three times independently (ns, not significantly, * *p* < 0.05, and *** *p* < 0.001).

**Figure 2 ijms-23-03843-f002:**
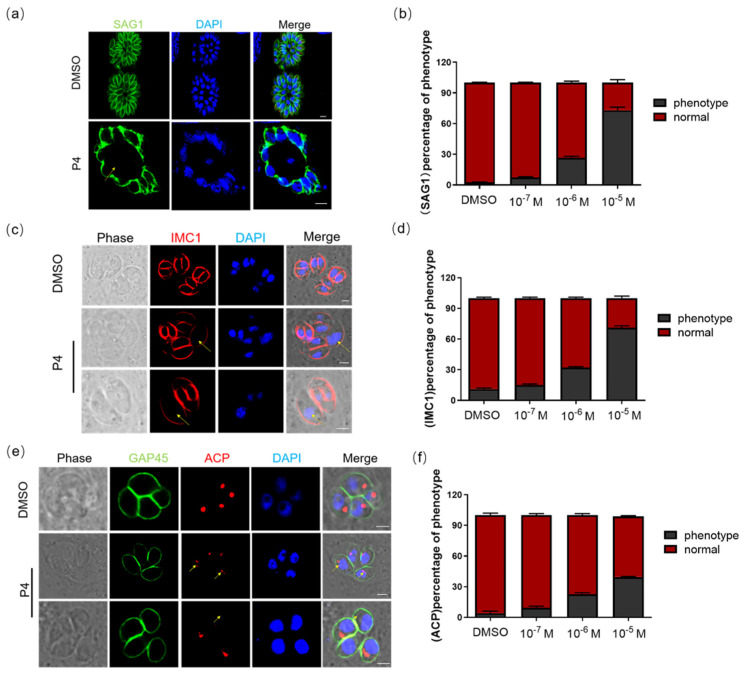
Progesterone can cause the abnormal morphology of *T. gondii*: (**a**,**b**) Cultured with different concentrations of P4 for 48 h and P4-induced parasites showed severe morphology defects (~8% under 10^−7^ M P4, ~27% under 10^−6^ M P4, and ~73% under 10^−5^ M P4). SAG1 (green), *T. gondii* surface protein, was used to calibrate the morphology of *T. gondii*. DAPI (blue) was used to stain the nuclei. Scale bar, 2 μm; (**c**) Cultured with different concentrations of progesterone for 48 h, parasites were determined by immunofluorescence. IMC1 (red) was used to label the parent and budding daughter cells. Scale bar, 2 μm; (**d**) Rate of asynchronous PVs: ~15% under 10^−7^ M P4, ~32% under 10^−6^ M P4, and ~75% under 10^−5^ M P4; (**e**,**f**) Progesterone resulted in abnormal division of the apicoplast (~10% under 10^−7^ M P4, ~23% under 10^−6^ M P4, and ~40% under 10^−5^ M P4). ACP (red), marker protein of apicoplast. GAP45 (green) was used to label parasites. Scale bar, 2 μm.

**Figure 3 ijms-23-03843-f003:**
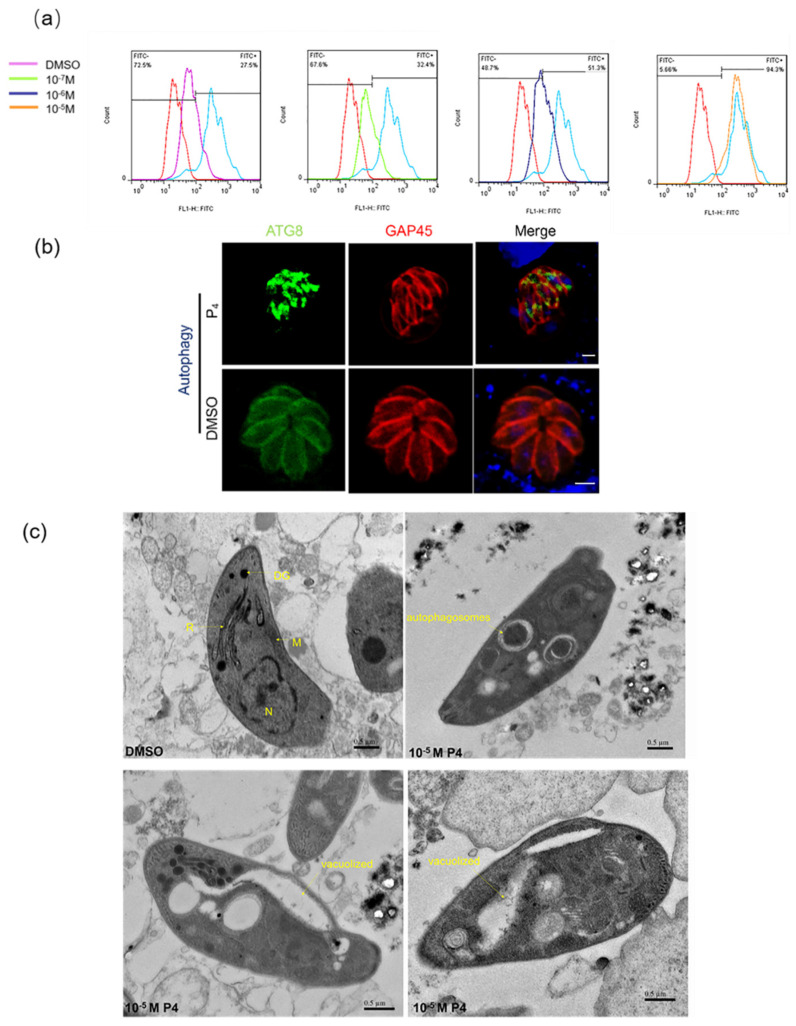
Progesterone induced autophagy in *T. gondii*: (**a**) Parasites were incubated with 10^−7^ M, 10^−6^ M, and 10^−5^ M progesterone for 3 h; then, parasites were stained with TUNEL and analyzed by flow cytometry; (**b**) Immunofluorescence results showed that 10^−5^ M progesterone promoted the accumulation of ATG8 in the cytoplasm of *T. gondii*. Scale bar, 2 μm; (**c**) Ultrastructural changes of *T. gondii* tachyzoites after being treated with P4. The well-preserved tachyzoite structures were maintained in the control group, including the nucleus (N), rhoptries (R), dense granules (DG), and mitochondrion (M). P4 treatment for 3 h caused many autophagosomes and vacuoles to emerge in the cytoplasm, as indicated by the arrows. Scale bar, 0.5 μm.

**Figure 4 ijms-23-03843-f004:**
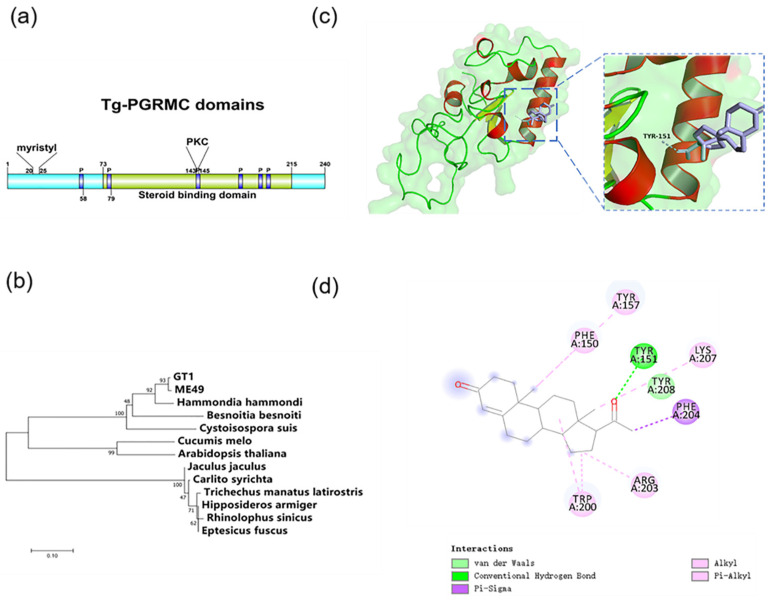
Characterization of structures, evolutionary relationships, and activities of TgPGRMC: (**a**) Conserved motifs of progesterone membrane receptor present in TgPGRMC; (**b**) Phylogenetic tree. Homologous alignments of *T. gondii* with the parasite, plant, and animal *pgrmc* genes are shown; (**c**,**d**) The template 4x8y (PDB sequence number) was chosen for modeling. According to the homologous family protein and pocket prediction, the interface pocket was determined, and then the molecular docking software AutoDock Vina was used to dock the three-dimensional structure of the protein with progesterone and to analyze the docking score and interaction residues.

**Figure 5 ijms-23-03843-f005:**
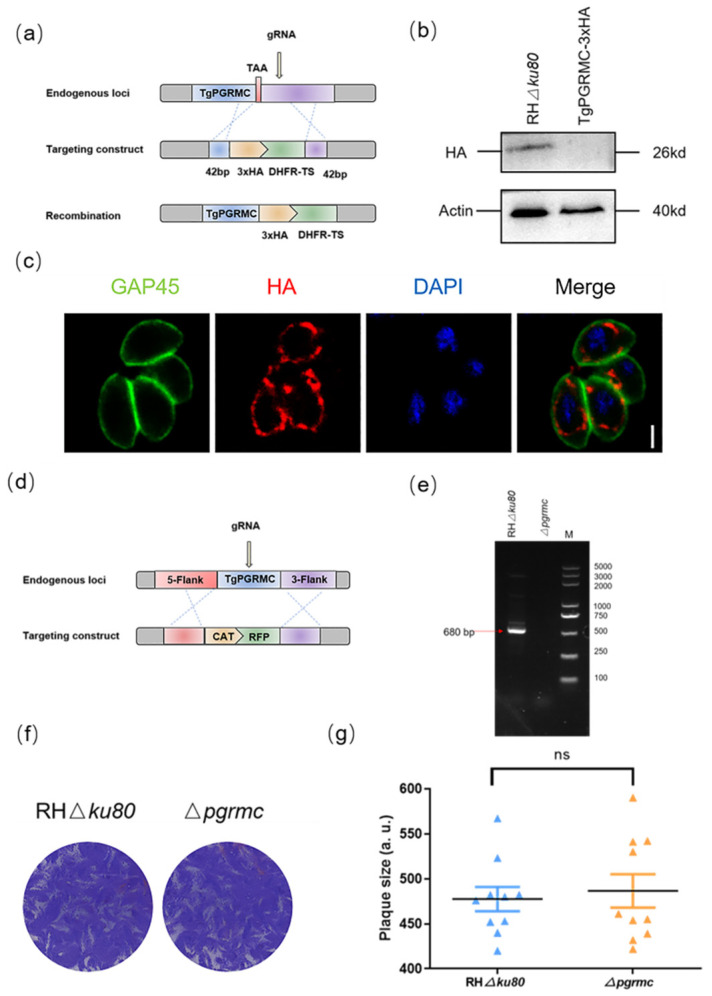
Phenotypic test of *pgrmc* gene knockout strain: (**a**) Strategy for construction of TgPGRMC-HA strain; (**b**) Western blot confirmed the expression of HA-tagged TgPGRMC in parasites. Actin was used as a control; (**c**) Immunofluorescence assays determined the localization of TgPGRMC. Parasites were labeled with mouse anti-GAP45 (green), mouse anti-HA (red), and DAPI (blue). Scale bar, 2 μm; (**d**) Strategy for constructing Δ*pgrmc* strain; (**e**) PCR analysis confirmed the *pgrmc* gene knockout; (**f**,**g**) Plaque formation of RHΔ*ku80* and Δ*pgrmc* strains for 7 days. Plaque assays were carried out three times independently (ns, not significantly).

**Figure 6 ijms-23-03843-f006:**
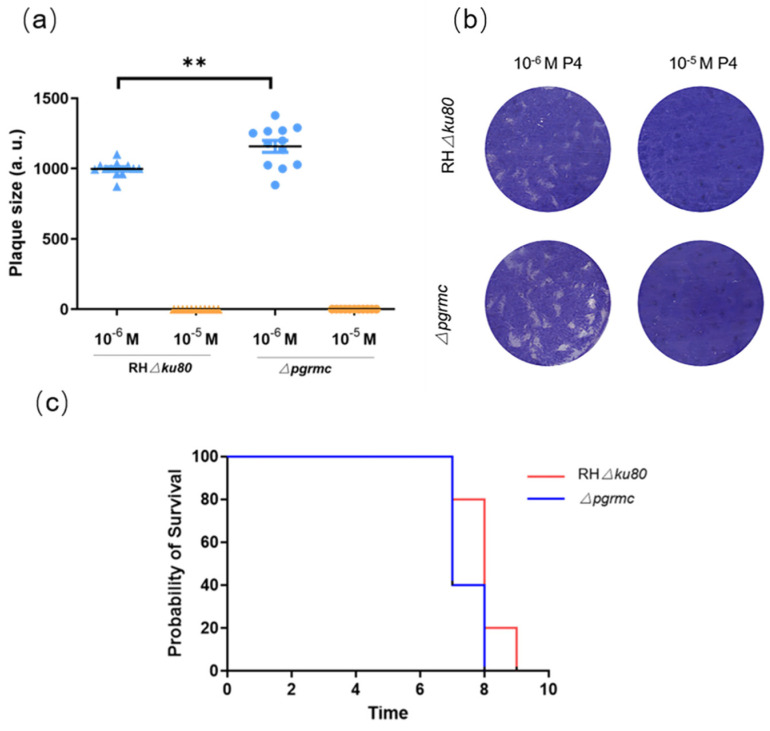
TgPGRMC is related to the inhibition effect of P4 on *T. gondii*: (**a**,**b**) Plaque assay. The survival of the wild-type strain and the *pgrmc* gene knockout strain under P4 culture conditions. Δ*pgrmc* strain conferred resistance to 10^−6^ M P4 inhibitory effects (** *p* < 0.01); (**c**) Survival curve of BALB/c mice intraperitoneally infected with 100 tachyzoites of RHΔ*ku80* strain or *Δpgrmc* strain.

## Data Availability

All datasets generated for this study are included in the manuscript/Appendix A.

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
