# Peer review of "Progesterone Can Directly Inhibit the Life Activities of Toxoplasma gondii In Vitro through the Progesterone Receptor Membrane Component (PGRMC)"

_ijms, 2022, doi:10.3390/ijms23073843_

Round 1

Reviewer 1 Report

The manuscript entitled “Progesterone can directly inhibit the life activities of Toxoplasma gondii in vitro through the progesterone receptor membrane component” was reviewed.

This is a very interesting study dealing with the pathogenesis and immune responses to Toxoplasma gondii infection.

 The introduction provides sufficient background and includes all relevant references. The study design is appropriate for the scope of the study and follows all relevant guidelines. The methodology appears to be well-grounded and it is adequately described. In most parts of the manuscript, English language and style are perfect and only minor spell check is required, there are some sentences that should be re-written as they are hard to follow:

Lines 46-47: please change the verb “prefer”

Lines 49-50: the immune reaction is “produced” by the maternal organism. Please rephrase

Lines 64-65: parasites are not susceptible to hosts. Please rephrase.

Lines 70-72: At the end of the introduction the aim of the study should be presented without any result. Please modified the text accordingly

Graphics are perfect regarding the content. However, I would recommend to the authors to increase the printing quality mainly figures 1,2,4 and/or increase the size of letters because it is difficult to read

Author Response

对审稿人 1 条评论的回复

综述了题为"黄体酮可以通过黄体酮受体膜成分在体外直接抑制弓形虫的生命活动"的手稿。

这是一项非常有趣的研究,涉及弓形虫感染的发病机制和免疫反应。

导言提供了足够的背景,并包括所有相关的参考资料。研究设计适合研究范围,并遵循所有相关指南。该方法似乎有充分的理由,并作了充分的说明。在手稿的大部分内容中,英语语言和风格是完美的,只需要少量的拼写检查,有些句子应该重写,因为它们很难理解:

第46-47行:请更改动词"首选"

回应:感谢您的建议!我们纠正了修订稿中的错误(第47-48行)。

第49-50行:免疫反应由母体生物体"产生"。请改写

回应:感谢您的建议!我们纠正了修订后的手稿中的错误(第50-51行)。

第64-65行:寄生虫不易受宿主影响。请改写。

回应:感谢您的建议!我们纠正了修订稿(第66-68行)中的错误。

第70-72行:在引言结束时,应提出研究的目的,但没有任何结果。请相应地修改文本

回应:感谢您的建议!我们在修订后的手稿中相应地修改了文本(第74-76行)。

图形在内容方面是完美的。但是,我建议作者提高印刷质量,主要是数字1,2,4和/或增加信件的大小,因为它难以阅读

回应:感谢您的建议!我们增加了字母的大小。但我们不确定它是否会得到改善。我们怀疑Word文件中的图形质量会降低。

Reviewer 2 Report

The authors have focused in their article on analyzing the effect of the hormone progesterone with respect to Toxoplasma gondii infection.
The introduction contains adequate information to expose the item under study. The results have been presented concisely and clearly, congratulating the authors for the methodology followed. Discussion clearly collects the results and compares it with other studies.
In materials, only comment that the "Electron microscopy" section should reflect the capture conditions, distance, vacuum... As in the images reflected in the  results section

Author Response

Response to Reviewer 2 Comments

The authors have focused in their article on analyzing the effect of the hormone progesterone with respect to Toxoplasma gondii infection.
The introduction contains adequate information to expose the item under study. The results have been presented concisely and clearly, congratulating the authors for the methodology followed. Discussion clearly collects the results and compares it with other studies.
In materials, only comment that the "Electron microscopy" section should reflect the capture conditions, distance, vacuum... As in the images reflected in the results section

Response: Thanks for your suggestion! We describe the material method in more detail in the revised (lines 360-362). Our title "Electron microscopy" may be misleading, we are actually using transmission electron microscopy, so no distance or vacuum information can be provided.
